# Exploring Grammatical Gender Agreement in Russian Learners of Greek: An Eye-Tracking Study

**Alexandros Tantos** [1,*] , **Nikolaos Amvrazis** [1] , **Konstantinos Angelou** [2] and **Kosmas Kosmidis** [2]

1   Department of Linguistics, Aristotle University of Thessaloniki, 541 24 Thessaloniki, Greece;
    amvrazis@lit.auth.gr
2   Department of Physics, Aristotle University of Thessaloniki, 541 24 Thessaloniki, Greece;
    kaggelou@auth.gr (K.A.); kosmask@auth.gr (K.K.)
*   Correspondence: alextantos@lit.auth.gr

**Abstract:** This study investigates the acquisition of grammatical gender in Russian learners of Greek. Agreement in Determiner-Noun (Det-N) and Adjective-Noun (Adj-N) dependencies is explored through eye-tracking registration in a reading-based design. Twenty-four intermediate learners read short texts embedded with agreement violations, and then responded to a comprehension task. The study implemented a two-level triangulation by drawing its stimuli from the Greek Learner Corpus II (GLCII) and contrasting, at a first level, the findings with comparable offline data that were previously obtained from the same corpus. The second level entailed a contrast between the online evidence and the offline data that were collected through a post-reading questionnaire right after the online eye-tracking session. This questionnaire explored whether longer fixations on agreement violations are associated with explicit awareness of the study's focus. To anticipate the outcome of the study, the gender agreement data suggests that the abstract gender feature is present in the developing grammar of Russian learners of Greek. Moreover, the participants seem to effectively deal with the syntactic computations underlying nominal agreement, though efficacy varies across the structures that have been examined. Apart from this, certain suggestions are made considering the research paradigm followed.

**Keywords:** grammatical gender; agreement; eye-tracking; triangulation; SLA

## 1. Introduction

Choosing gender as the focal point of this study is motivated by its scope across grammatical levels, encompassing morpho-phonology, core semantic features, and lexicon. From this perspective, elucidating the acquisition of gender agreement provides insight into multiple levels of linguistic representation (Polinsky 2008, p. 41). Moreover, acquiring grammatical gender poses challenges for many second language (L2) learners even at the advanced level of interlanguage (IL) and after prolonged exposure to the L2 (e.g., Carroll 1989; De Garavito and White 2002; Tsimpli et al. 2007; Franceschina 2005).

Considering the specific pair of languages in the present study (Russian and Greek) an interesting overlap is observed in the syntactic entities that agree with the noun, since Russian employs adjectives but not determiners[1] (see Section 2.2). This partial overlap facilitates exploration of the processing mechanism underpinning the computations for morpho-phonological agreement, with particular implications for the role of the L1.

Therefore, this work aims to contribute to the ongoing debate concerning the availability of functional features in SLA. It is postulated adult native speakers of Russian will only be partially facilitated in aligning modifiers (i.e., adjectives but not determiners) with nouns in Greek. In formal models that subscribe to defective grammatical representations in the L2, such alignment within the noun phrase (NP) operates based on parameterized functional features. These features are subject to maturational constraints and resist acquisition when absent in the L1 (e.g., Failed Functional Features Hypothesis; Hawkins and

Chan 1997). From this perspective, the L1 and L2 are fundamentally different in the way the abstract representations are encoded (e.g., Hawkins and Casillas 2008; Clahsen and Muysken 1986; Bley-Vroman 1990).

In turn, impaired representations may affect the way the parser is informed by grammar leading to shallow parsing (Clahsen and Felser 2006). In a more refined approach, formal features (i.e., the uninterpretable ones) not instantiated in the L1 are seen as the source of persistent problems in acquiring the relevant structures in the L2 (e.g., Tsimpli and Dimitrakopoulou 2007). However, these approaches have long been contrasted with competing accounts (e.g., Full Transfer Full Access Hypothesis; Schwartz and Sprouse 1996) in which the L1 grammar is assumed to be the initial state of the learners' IL. In these models, resetting is possible as UG remains available to the adult L2 learner (L2er). They posit that L1 and L2 grammars are not qualitatively different and processing restrictions appear to be the source of non-target performance as the learner occasionally fails during online processing to connect abstract features to morphology (e.g., Feature Reassembly Hypothesis; Lardiere 1998, 2009).

The Feature Reassembly Hypothesis (FRH) suggests that L2 development requires restructuring of the L2 features, a process initially informed by their instantiation in the L1 (Lee and Lardiere 2019). In this perspective, L2 acquisition is associated with the task of reassembling features on functional categories and lexical items. In the course of this task, achieving target-like reassembly seems challenging, and deviant performance is often attributed to processing restrictions that impede form-to-meaning mapping (see also, Missing Surface Inflectional Hypothesis; Prévost and White 2000). Importantly for this study, the FRH acknowledges that non-target reassembly of the L2 features is not limited to their different instantiation in the L1, and that processing restrictions are held even in language pairs that share the same feature settings.

It appears that any contribution to the above discussion should ideally incorporate data that enables the exploration of L2ers' representations in both offline and online processing. It is well established that syntactic resources are underutilized during L2 sentence comprehension (see Keating 2009 and related references therein). Therefore, having comparable evidence to test the processing of gender agreement in online and offline settings would enhance the investigation into the presence and the nature of abstract representations in the L2ers' grammar. In line with this approach, triangulation was applied with the online data of this study being comparable to naturalistic data previously examined. Specifically, errors in Det–N and Adj–N configurations as produced by L2ers in Greek Learner Corpus II (GLCII) were the critical stimuli that were employed in the violation detection paradigm employed in the eye-tracking registration of the present study.

Alongside triangulation, consideration was also given to the type of knowledge that the specific study would assess. The study of grammatical gender relies on exploring the abstract and implicit linguistic system of the L2er. Activation of conscious knowledge (i.e., explicit grammatical rules (see, DeKeyser 2003)) should be avoided during such an exploration. In fact, it has been observed that the detection of anomalies during a reading task could lead to a processing strategy (Keating and Jegerski 2015), whereby the learner resorts to the explicit grammatical rules that regulate the impaired structure. As a result, the ecological validity of the study, that is "the simulation of natural reading conditions" (Spinner et al. 2013, p. 389), is limited, as learners are not naturally engaged in processing input (see Godfroid 2020; Spino 2022). As Spino (2022, p. 184) notes "such concerns can be attenuated through triangulation", a methodology designed to test whether evidence from different measurements (e.g., experimental and naturalistic) are converging on the same phenomenon. The present study employs both within and between-method triangulation (Denzin 1989). The former involves the incorporation of different types of gender agreement (i.e., determiner–noun and adjective–noun) while the latter involves validation of the experimental data against corpus findings and a post-test questionnaire.

## 2. Syntactic Background: Gender

Following Corbett (1991), languages use either semantic or formal cues to classify nouns into genders. Thus, in Tamil, gender is semantically motivated, whereas in Russian and Greek it is subject to morpho-phonological cues. Grammatical gender in the latter languages functions as a nominal classifier in the lexicon, meaning that the noun enters the numeration pre-specified with a gender feature (Chomsky 1995). This feature is interpretable at the Logical Form (LF) (Carstens 2000).

Gender agreement is the reflection of the grammatical gender at the syntactic level. At this level, and within the minimalist framework (Chomsky 1995; Carstens 2000), the interpretable gender feature of the noun is assumed to check the uninterpretable gender feature on agreeing categories (e.g., determiners, adjectives, pronouns). In this view, the uninterpretable features are responsible for feature alignment within the noun phrase (NP) or even across the sentence. From the learner's standpoint, the morphological realization of the agreeing categories depends on the respective morphological realization of the target noun (e.g., **o**-$_{MASC}$/i-$_{FEM}$/to$_{NEUT}$ fov**os**-$_{MASC}$ "*the fear*") implying that the learner must exploit morphological and semantic cues in order to select the target value (Carroll 1989).

### 2.1. Gender in Greek

Greek exhibits a tripartite division of nouns into masculine, feminine, and neuter gender (Seileb 1958; Holton et al. 2012, among others). This gender assignment is largely arbitrary, though semantic properties, such as animacy, have a tendency to favor the assignment of animate nouns to the masculine or feminine gender (Tsimpli and Hulk 2013; Setatos 1998; Christofidou 2003).

Ralli (2002) proposed that in Greek, gender is encoded either in the noun root (1a) or in the noun suffix (1b).

| (1) | | Masculine | Feminine |
|-----|-----|-----------|----------|
| | a. | ji-os | kor-i |
| | | son-$_{NOM}$ | daughter-$_{NOM}$ |
| | b. | mathit-is | mathitri-a |
| | | student-$_{NOM}$ | student-$_{NOM}$ |

This view is followed by Tsimpli and Hulk (2013), who claim that gender assignment follows either the lexical (1a) or the morpho-phonological (1b) route, with the path being determined by the predictive effect of the root or the suffix. For instance, even though masculine, feminine and neuter nouns can take the suffix "os", adult native speakers of Greek are biased by the prototypicality of the marker (Anastasiadi-Symeonidi and Cheila-Markopoulou 2003) and attribute a novel noun with that suffix to the masculine gender (Varlokosta 2011). When it comes to developing L2 grammar, it is expected that the predictive values will become accessible and operative for the learner once a certain amount of vocabulary has been acquired (Mastropavlou and Tsimpli 2011).

Critically, the morpho-phonological cues—lexical or affixal—are not reliable sources for gender attribution:

| (2) | | or-os$_{-MASC/NEUT}$ | | | | | |
|-----|-----|------|--------|-----|-----|------|---------------|
| | a. | o | or-os | b. | | to | or-os |
| | | the | term-$_{MASC}$ | | | the | mountain-$_{NEUT}$ |

As shown in (2), grammatical gender, an inherent feature of nouns, may not be accessed unless the speaker recognizes the gender in the Det–N (as in 2a and 2b) or the Adj–N concords. This is because in (2) both the suffix and the noun stem are phonologically underspecified for grammatical gender (Ralli 2003, p. 82). Consequently, any gender information stored in the mental lexicon alongside the noun stem, or the suffix cannot be activated.

Likewise, the gender of the nouns in (3) is available through the words associated:

|  | | Det–Noun | | Adj–Noun | |
|---|---|---|---|---|---|
| (3) | a. | o | kikn-os | omorf-os | kikn-os |
|  |  | the-MASC | swan-MASC | beautiful-MASC | swan-MASC |
|  | b. | i | perioδ-os | omorf-i | perioδ-os |
|  |  | the-FEM | period-FEM | beautiful-FEM | period-FEM |
|  | c. | to | mer-os | omorf-o | mer-os |
|  |  | the-NEUT | place-NEUT | beautiful-NEUT | place-NEUT |

Example (3a) exhibits a prototypical masculine-ending noun with the suffix (-os) in agreement with the determiner and adjective. The same -os suffix is observed in (3b) and (3c). Thus, gender assignment is determined only through agreement with the article or the adjective. In cases like those in (3), the learner is challenged with the task of internalizing the grammatical gender of a predominantly formal system: in most cases, they must learn the determiner–noun pairings for each noun[2]. To achieve target-like acquisition of grammatical gender, immersion in a sufficient volume of input is, thus, necessary (Unsworth 2008, p. 367).

The following table presents the limited patterns that can be applied for gender assignment through a set of prototypical endings.

Even though the gender system as depicted in Table 1 appears to be straightforward, exceptions abound, and this limited set of gender patterns becomes puzzling when the case and number of the noun are taken into consideration. For example, deriving the (masculine) (singular) (accusative) form requires the omission of -s, resulting in syncretism with feminine and neuter endings. Therefore, the nominal suffixes are not reliable gender markers in Greek. Importantly for this study, definite articles (i.e., *o, i, to*) can be quite useful in determining the gender of the noun that follows.

**Table 1.** Suffix–gender correspondences for prototypical endings in Greek.

| Gender Assignment in Greek | | | | | |
|---|---|---|---|---|---|
| **Masculine** | | **Feminine** | | **Neuter** | |
| **Orthographic** | **Phonetic** | **Orthographic** | **Phonetic** | **Orthographic** | **Phonetic** |
| ο καιρ ός [o çeros] | [os] | η πόρτ α [i porta] | [a] | το βιβλί ο [to vivlio] | [o] |
| ο μήν ας [o minas] | [as] | η αυλ ή [i avli] | [i] | το χαρτ ί [to xarti] | [i] |
| ο δείκτ ης [o δiktis] | [is] |  |  | το θέ μα [to θema] | [ma] |

Adjectives agree with the noun in both attributive and predicative position. Agreement is realized with inflectional suffixes which in some cases are phonologically aligned with the corresponding suffixes of the noun (see examples 4a, 4b, 4c).

| (4) | a. | kal-os | rol-os |
|---|---|---|---|
|  |  | good-MASC/NOM/SG | role-MASC/NOM/SG |
|  |  | kal-os | pinak-as |
|  |  | good-MASC/NOM/SG | painting-MASC/NOM/SG |
|  | b. | kal-i | efh-i |
|  |  | good-FEM/NOM/SG | wish-FEM/NOM/SG |
|  |  | kal-i | teni-a |
|  |  | good-FEM/NOM/SG | movie-FEM/NOM/SG |
|  | c. | kal-o | sxoli-o |
|  |  | good-NEUT/NOM/SG | comment-NEUT/NOM/SG |
|  |  | kal-o | taksiδ-i |
|  |  | good-NEUT/NOM/SG | trip-NEUT/NOM/SG |

Phonological agreement such as in (4a, 4b, 4c) extends to plural as well, thus supporting suggestions (e.g., Agathopoulou et al. 2008; Tsimpli et al. 2007) that the repetition of the noun suffix onto the adjective (e.g., *statheres ipolojistes* instead of *statheri ipolojistes* "*desktop computers*") is a strategy employed by the learner as a reflection of the phonological matching in the input.

### 2.2. Gender in Russian

Russian shares the same three-way gender classification. Gender assignment is formally motivated except for a limited semantic core which is operative for nouns bearing (+animate) features (see Corbett 1991; Comrie et al. 1996). Aside from gender, inflectional morphology encodes case and number. Notably for this research, gender concord is only expressed through adjectives, given the absence of determiners in Russian. Thus, gender attribution is determined by the declensional class once the learner notices the suffix of the unmarked (nominative) case (Polinsky 2008, p. 42). Regarding agreement, masculine adjectives bear a distinct suffix while the female–neuter distinction is realized through phonological means (i.e., stress). Adjectives with the stress on the ending maintain the distinction, while those with the stress on the stem are ambiguous regarding the marking of female or neuter. Table 2 presents noun and adjective endings (nominative, singular) according to gender values.

**Table 2.** Suffix–gender correspondences of nouns and adjectives in Russian (adapted from Hopp and Lemmerth 2018).

| Suffix Distribution across Gender in Russian | | |
|---|---|---|
| | Noun Suffixes | Adjective Suffixes |
| Masculine | -a, Ø, -ь | -ый |
| Feminine | -a, -ь | -ая |
| Neuter | -о, -я | -ое |

The above system extends to four declensional classes (however see Parker and Sims 2020 for a detailed overview), each manifesting six cases and two numbers. The rich paradigm along with the phonological properties of the noun or the adjective contribute to a relatively opaque system for gender assignment (e.g., Popova 1973; Polinsky 2008; Karpava 2021; Hopp and Lemmerth 2018).

In conclusion, Greek and Russian exhibit a common tripartite gender system. Gender is determined by formal criteria, with semantic distinction reserved for animate nouns. Despite this similarity, the two languages diverge in terms of the agreement categories that make up the noun phrase, since Russian does not utilize determiners, resulting in an overlap with Greek only in adjective–noun concord. Grounding these considerations in the task of learning we can assume that a Russian speaker of Greek should acquire not only the inherent lexical feature for each noun but also the respective uninterpretable gender feature of determiners (Ds) which is parameterized in the two languages. Moreover, the inherent property of lexical classification along with the derivative one of gender attribution on agreeing categories should be at the learners' disposal during the course of language use. In other words, agreement errors of Russian-speaking learners of Greek should be interpreted as a failure at:

A.　the lexical level, meaning that the learner has not assigned the noun to the proper gender category,

B.　the syntactic level, meaning that the learner has no access to the uninterpretable features of the determiner and the adjective even though they are aware of the correct gender value of the noun,

C.　the processing level, meaning that the learner fails to implement the proper computations and reassemble the features of lexical and grammatical gender due to

processing restrictions even though the abstract features may be shared by the L1 and L2.

Within the Generative tradition, scenarios A and B are accounted for by claiming the general issue of the accessibility of UG and the respective nature of the L1 and L2 (e.g., Clahsen and Muysken 1986; Bley-Vroman 1990; Beck 1998; Hawkins et al. 2004; Franceschina 2005). The third one is better accommodated within a framework that discusses the L2ers' performance on grammatical gender in the context of the limitations imposed by real-time use (e.g., Lardiere 1998; Prévost and White 2000; Clahsen and Felser 2006). This study provides evidence for the presence of the above formal features at both the lexical and syntactic levels in the developing grammars of Greek. Moreover, it explores the impact of online processing on the computations that are involved in nominal agreement.

### 3. Background on Gender Agreement and Online Processing

Despite both Greek and Russian featuring grammatical gender, they employ different agreement categories, leading to only a partial overlap at the syntactic level. Syntactic congruency between L1 and L2 has been shown to influence the processing of gender agreement violations (e.g., *de$_{com}$ kleine kind$_{NEUT}$) in ERP studies (e.g., Sabourin and Stowe 2008). Late acquirers whose L1 does not realize gender, responded differently to gender violations compared to the control group. Syntactic congruency extends beyond the availability, or the type of grammatical categories involved in agreement. It also encompasses the positions that allow gender marking and the realization of the φ-features (i.e., number and case). For instance, in an ERP study, German learners of French showed no sensitivity to gender violations when the adjective followed the noun (i.e., DNA vs. DAN orders). Foucart and Frenck-Mestre (2011) attributed this finding to the influence of the learners' L1 (i.e., German), where postnominal adjectives are not an option.

Therefore, both the placement of the categories within the agreement concord and the availability of inflectional cues impacts the extent to which the L1 affects processing of gender violation in the L2. The overlap between the φ-features instantiated in L1 and L2 has also been shown to modulate the L2ers' sensitivity to agreement violations. For example, an ERP study showed that English learners of Spanish could detect gender violations in the L2, but not number agreement violations in the noun phrase. This discrepancy in responses to gender and number agreement violations was attributed to the absence of number agreement between articles and nouns in English (Tokowicz and MacWhinney 2005).

Besides the specific research questions, the above electrophysiological comprehension studies are congruent in their observation that L2ers are sensitive to gender agreement violations even though a native-like pattern is subject to various factors. However, a critical imbalance is reported when Det–N violations are contrasted to Adj–N violations. The latter trigger an attenuated ERP signature (Foucart and Frenck-Mestre 2011; Dowens et al. 2011), suggesting that Adj–N dependencies are subject to shallow processing.

Processing of grammatical gender agreement has also been the focus of studies in which eye-tracking is implemented. The visual world paradigm and text-based studies are the most common experimental techniques used (for a review see Spino 2022). In the former, the researcher can explore the role of the syntactic overlap between the L1 and the L2 and measure the anticipatory fixations that are auditorily triggered by the morphosyntactic stimuli preceding the noun (i.e., the gender cues upon the determiner or the adjective). A common observation in a number of studies within this paradigm (e.g., Lew-Williams and Fernald 2010; Dussias et al. 2013; Grüter et al. 2012; Hopp 2013) is the reduced sensitivity to gender markers by L2ers whose L1 does not feature grammatical gender. These L2ers could not predict the unfolding noun on the basis of the preceding gender cues, a pattern that was not replicated by L2ers whose L1 instantiated gender (but see counter evidence in Martin et al. 2013 or Grüter et al. 2016). Hopp and Lemmerth (2018) explored the conditions that might interfere with predictive processing of gender and confirmed their hypotheses about the role of the L1 settings and the proficiency level in L2. Syntactic congruency

between Russian and German, particularly as manifested through suffixes upon nouns and adjectives, combined with a high proficiency level, facilitated predictive reading and led to target-like L2 gender processing.

In text-based eye-tracking registration the participants are exposed to stimuli containing grammatical violations. Keating (2009) explored the processing of gender agreement by implementing the violation detection paradigm. The locus of nominal agreement (DP, VP, subordinate clause) and the proficiency level were the main factors of the study. Advanced English-speaking learners of Spanish exhibited longer fixations for agreement gender violations but only for the within-DP condition. The results were taken as an indication that gender agreement is ultimately attained while shallow processing (Clahsen and Felser 2006) of morphological units emerges when the structural distance increases. Within the same paradigm, the design employed by Spino (2022) is relevant to the scope of this study. The researcher employed a triangulation approach to increase confidence about the type of knowledge—implicit or explicit—that was activated in English-speaking learners of Spanish. This was performed by combining the results from an eye-tracking violation detection paradigm with post-reading questionnaire data. The sentences presented in the study contained agreement violations in Det–N and Adj–N dependencies. The results of this study indicate that the participants were able to detect Det–N violations, a structure which was further evidenced in the post-reading questionnaire. Additionally, the post-reading questionnaire suggested that explicit knowledge had been activated, emphasizing the importance of a well-structured experimental design that draws upon multiple measurements for triangulation.

To sum up, a range of variables, such as the presence of grammatical gender in the L1, the overlap of the related agreeing categories, the locus of agreement, and the proficiency level, have been identified as key factors in the online processing of gender agreement. Even though it is virtually impossible to find common ground among all those studies, the general picture suggests that L2ers can acquire gender even when their L1 does not encode the respective features, with Adj–N structures being more likely to trigger erroneous performance especially in real-time language use.

Inspired by the existing body of work, the primary motivation of this study is to contribute to the field by presenting data gathered through both online and offline methods. The language pairing in our study permits an in-depth exploration of the syntactic mechanisms involved in agreement concords. As highlighted in this section, there is a lack of consensus regarding the impact of the realization of these mechanisms in L1 on L2 acquisition. Furthermore, the triangulation approach adopted in our research is designed to eliminate potential confounding variables.

The present study implements the violation detection paradigm in a text-reading design. The main advantage of the text-based study is that it emphasizes the contribution of both, morphophonological cues of the noun and morphosyntactic cues of the agreeing categories. Unlike the visual world paradigm, the focus extends from the facilitative role of the morphosyntactic cues to the impact of the lexical gender as the latter is formally encoded on the noun (Spinner et al. 2013). At the same time the experimental design addresses the challenge of filtering out the explicit knowledge when measuring the learners' underlying linguistic competence.

## 4. Study

### 4.1. Participants

Twenty-four adult learners of Greek participated in the study. All these learners were monolingually raised Russian speakers. At the time of the experiment, they had been living in Greece for an average of 14 months and were taking Greek courses at the Aristotle University of Thessaloniki. Their proficiency level was B1 ($n = 12$) and B2 ($n = 12$) and was assessed through:

a.    a placement test that was taken at the beginning of the course,

b.  qualitative data, i.e., an interview with the teacher who indicated the participants that met the specific criteria[3].

In this experiment, 80 h of in-class participation was completed by the participants prior to data collection. The participants' profile is presented in Table 3.

**Table 3.** Profile of the participants' pool.

| Participants' Profile | |
|---|---|
| Participants | 24 |
| Country of origin | Russia |
| Female | 11 |
| Mean Age | 30.8 (SD: 9.8) |
| Mean length of exposure to Greek | 14 months |

*4.2. Preliminary Considerations on the Design*

The current study uses the violation detection paradigm through a reading-comprehension mode of eye-tracking registration. Exposing L2ers to gender agreement anomalies could certainly activate explicit rule-based knowledge, thus increasing the risk of measuring unwanted resources. To tackle this inherent flaw of the specific paradigm we attempt to ensure its ecological validity (Spinner et al. 2013) by:

a.  contextualizing the critical stimuli (i.e., areas of interest, AOIs) in a short passage, rather than in a sentence,
b.  providing reading comprehension questions that required a careful consideration of the passage,
c.  coupling the eye-tracking with qualitative data (post-test questionnaire).

As mentioned in Section 1, the current study is data-driven in that the stimuli pool used in the eye-tracking registration was informed by the agreement violations found in a corpus study (Tantos and Amvrazis forthcoming; see also stimuli design in López-Beltrán and Dussias forthcoming). Therefore, the study's design should enable a comparable setting between the context where the violations occurred (i.e., in GLCII) and the context where the violations would be processed, thereby neutralizing the context as a possible confounding factor. To ensure a comparable set of data, certain compromises had to be made. The following have been adopted in the present study:

a.  The nouns used as critical stimuli were chosen based on the noun suffixes that elicited a significant number of agreement violations in the corpus. Hence, nouns are equally dispersed across morphemes (see Section 4.3) and not across the formal features (i.e., gender, number, case).
b.  Relating to the above, the stimuli are not balanced for gender, number, or case. For example, it is not meaningful in the current study to counterbalance the violation in $pol\text{-}es_{FEM/PL}$ $min\text{-}es_{MASC/PL}$, a common one in the corpus, with stimuli in the singular form of the same gender and case. Instead, a proportionate amount of Det–N and Adj–N agreement violations across morphological categories were included in the stimuli pool. Moreover, the above agreement settings were balanced for phonological alignment (see Section 2.1) resulting in the two variables we explore in the study.

The following parts of this section illustrate the way the design of the current study is shaped by the preceding considerations.

*4.3. Materials and Methods*

4.3.1. Eye-Tracking Stimuli

Data from the Greek Learner Corpus II[4] (GLCII), the largest available online corpus of L2 Greek, were used for preparing the items in this experiment. GLCII is annotated for grammatical gender agreement with determiners and adjectives. To triangulate naturalistic data with experimental data, 208 agreement violations from 155 productions from Russian

learners of the same proficiency level were collected from GLCII. An analysis of these errors (Tantos and Amvrazis forthcoming) across the syntactic structure (i.e., Det–N and Adj–N) revealed that:

a.    Adj–N agreement is much more challenging than Det–N agreement, inflicting twice as many errors as Det–N agreement. However, note that the proportion of errors in the total production of the agreement structures was low (13% and 7%, respectively) indicating that the agreement computations were operating in those subjects.

b.    Phonological agreement (see example 4, Section 2.1) is a pattern that significantly affected the error rate in both structures.

c.    A Det–N gender violation (e.g., tin$_{FEM}$ planiti$_{MASC}$), even though it is often taken as evidence for gender assignment (e.g., Lew-Williams and Fernald 2010) it could also be attributed to impaired computations in the course of agreement (e.g., Montrul et al. 2008). Disentangling gender assignment from gender agreement is not a trivial task in corpus annotation (see Tantos and Amvrazis 2022) which could be substantially informed by triangulation.

A total of 20 nouns (×2 blocks) from the above pool, standardized for length (two to three syllables, balanced) and suffix,[5] were the target items that appeared in Det–N (ten items) and Adj–N (ten items) configurations. In each structure, half of the items were phonologically aligned (e.g., **ti**$_{FEM}$ frah**ti**$_{MASC}$, mikri$_{FEM}$ frah**ti**$_{MASC}$) and the rest contained violations with phonologically mismatching morphemes (e.g., **ti**$_{FEM}$ δe**ma**$_{NEUT}$, mikri$_{FEM}$ δe**ma**$_{NEUT}$). In the Det–N condition, definite articles were employed, whilst adjectives of two to three syllables were evenly distributed in the Adj–N condition. The selection of adjectives was based on the same pool of 208 gender violations from the GLCII. The adjectives were used in attributive settings (e.g., kalos mathitis "good student") with predicates (e.g., o mathitis ine kalos "the student is good") being excluded in the study. All violations were located within NPs, with equal dispersal in subject, object and adverbial[6] position (prepositional phrase).

In the study, 80 nouns (including the 20 critical and 20 control items) were embedded in short texts of 78–85 words, with each text containing either 10 or 11 sentences. The ratio of critical to non-critical items was 25%[7]. Two blocks of four texts were created, with each text containing five violations of a certain condition along with five noun phrases (NPs) in the control condition, presented in both Det–N and Adj–N configurations. Each text was followed by a comprehension task in a multiple-choice format (three alternative answers) (see Appendix A).

Participants were randomly assigned to one of the two blocks and the conditions were counterbalanced to ensure that each participant read either the Det–N version of the doublet (e.g., in non-phonological agreement *to epohi*) or the Adj–N condition (e.g., *neo epohi*). This is exemplified in the table below.

The order of the texts in each block was randomized to ensure that the conditions were not presented in the same order for all participants. Table 4 shows that although each noun was featured in two conditions, the critical items were encountered only once by the participants.

Participants were familiarized with the process by following a practice session before the commencement of the experiment. This session included a text and the associated comprehension task. A criterion of 80% (minimum) accuracy in responses was established to assess whether participants had properly read the texts and complied with the instructions. All participants surpassed this threshold, achieving an average accuracy of 88%[8].

**Table 4.** Counterbalanced conditions in the two blocks.

| Study's Conditions | | | |
|---|---|---|---|
| **1st Block (12 Participants)** | | **2nd Block (12 Participants)** | |
| Det–N (non phonological) | $to_{NEUT/NOM/SG}$ $epohi_{FEM/NOM/SG}$ *"the season"* | Adj–N (non phonological) | $neo_{NEUT/NOM/SG}$ $epohi_{FEM/NOM/SG}$ *"new season"* |
| Det–N (phonological) | $o_{MASC/NOM/SG}$ $meros_{NEUT/NOM/SG}$ *"the place"* | Adj–N (phonological) | $mikros_{MASC/NOM/SG}$ $meros_{NEUT/NOM/SG}$ *"small place"* |
| Adj–N (non phonological) | $kalus_{MASC/ACC/PL}$ $times_{FEM/ACC/PL}$ *"good prices"* | Det–N (non phonological) | $tus_{MASC/ACC/PL}$ $times_{FEM/ACC/PL}$ *"the prices"* |
| Adj–N (phonological) | $kala_{NEUT/ACC/PL}$ $mina_{MASC/ACC/SG}$ *"good month"* | Det–N (phonological) | $ta_{NEUT/ACC/PL}$ $mina_{MASC/ACC/SG}$ *"the month"* |

### 4.3.2. Post-Reading Questionnaire

With this specific tool we elicited qualitative data to measure the chances of conscious detection of the violations in the study, thus implementing between-method triangulation. The participants answered the following questions: (adapted from Spino 2022, p. 197)

*Q1. Did you notice anything strange about the short texts you read? If so, what?*

*Q2. Were there any grammatical errors in the short texts you read? Please check.*

*Yes                                            No*

*Q3. What types of grammatical errors did you notice? Please list all the errors you remember and provide examples when possible.*

*Q4. Please check off the types of errors you noticed in the texts. If you are unsure what something is, please ask the researcher.*

    *Determiner–noun agreement violations*
    *Spelling mistakes*
    *Subject–verb agreement violations*
    *Illicit use of certain nouns*
    *Adjective–noun agreement violations*
    *Tense–adverb violations*
    *Illicit use of fonts*

*Q5. What percentage of the passages you read contained errors?*

A cutoff point was established for the second question in order to terminate the process if the response was "no", as questions 3–5 require that the participant has detected errors (i.e., a "yes" response). Following the original format of the tool, participants were only presented with one question per page, thereby preventing any influence from subsequent questions. It is worth noting that the researcher was present during the post-reading questionnaire session to provide clarifications as needed.

### 4.3.3. Vocabulary and Gender Assignment Post-Test

As previously mentioned, this study relied on the written productions of Russian-speaking learners of Greek, recorded in the GLC, whose proficiency level matched that of the participants in this study. We selected the critical nouns and adjectives based on the criteria outlined in Section 4.3. We further verified them by consulting the learners' teachers about their comprehension of the specific texts and critical vocabulary. To ensure the validity of the stimuli, we conducted a pilot test with five students from the same

courses. These students ranked the familiarity of the nouns on a scale of 1 to 3 (see Spinner et al. 2013).

The items that were selected—with a minimum familiarity score of 2.79—and used in the experiment have also been included in the post-test. The items that were selected -with a minimum familiarity score of 2.79- and used in the experiment have also been included in the post-test. In this test, the participants had to translate the nouns into Russian and assign each noun a gender through the use of an article. This would allow us to rule out the possibility that longer fixations are due to incorrect lexical assignment and focus on the computations at the level of the agreeing categories.

### 4.4. Procedure and Screen Layout

In a laboratory setting, the researcher first briefed the participant on the research context, including the experiment, and explained the terms of the consent form. The participants then read and signed the form, indicating their voluntary participation. The researcher instructed the participants to take the appropriate position in front of the eye-tracker and initiated the nine-point calibration of the hardware, ensuring the participant was positioned at between 55 and 65 cm.

A Tobii Pro Fusion eye tracker operating at 250 Hz was utilized to measure eye-movement registration. The eye-tracker was connected to a Hewlett-Packard laptop with a 15.6-inch screen, and Tobi Pro Lab (v 1.207) was the software used to generate the stimuli and manage the eye-tracker.

The stimuli appeared in black fonts (Verdana 44 pt.) on a light grey background. The line spacing was set to 2.0 to provide adequate space for the areas of interest (AOIs). The participants were instructed to read four texts, each followed by a comprehension question. They could advance to the next text by pressing the space button after answering each question. Participants were required to click the correct answer using the mouse. Once ready, they could initiate the practice session, which presented a text without violations and a subsequent comprehension question, by clicking the 'OK' button. The eye-tracking session lasted approximately 14 min.

The session progressed by having the participants complete the post-reading questionnaire, followed by the vocabulary and the gender assignment task. Finally, the background form, which was the same as the one used to select the participants' profiles in the GLCII[9], was completed. The entire procedure lasted approximately 40 min.

### 4.5. Research Questions

The current paper explores gender agreement in real-time comprehension by focusing on the role of morphosyntactic (determiners and adjectives) and morphophonological (phonological and non-phonological) agreement features, which were previously investigated through corpus-based data. Additionally, we considered how the design of our study might suppress the activation of explicit knowledge during eye-tracking registration. These key elements led to the expression of the following questions:

RQ1: Are the L2ers at the intermediate level sensitive to gender agreement violations during online processing?
RQ2: Is the Adj–N agreement less salient than the Det–N agreement?
RQ3: Are violations involving phonological agreement less salient, and hence less detectable by the L2ers?

Regarding RQ1, the prediction was that the L2ers would demonstrate sensitivity to agreement violations in both structures especially when considering the respective findings in the corpus study (see Section 4.3.1). Regarding RQ2, we anticipated that the sensitivity to Adj–N violations would be less pronounced than that to Det–N violations, a finding that would be aligned with evidence reported by a number of studies (e.g., De Garavito and White 2002; Franceschina 2005; Fernández-García 1999). The prediction for the RQ3 was that phonological agreement would make the violations less detectable with shorter fixations (e.g., Agathopoulou et al. 2008). A confirmation of the RQ2 and RQ3 would

also validate through triangulation, the corpus findings. Finally, we anticipated that the qualitative data selected by the post-reading questionnaire would show limited awareness of the study's focus. It is noteworthy that the participants in Spino's (2022) study reported high levels of awareness (approximately 60%). The different design of the critical stimuli in the two studies should be taken into account in this regard.

## 5. Results

This Section reports on the results in the order they were received. The session started with the eye-tracking registration. Then, the participants filled in the post-reading questionnaire which registered the levels of explicit knowledge that might be triggered by the experiment. Finally, the vocabulary test measured the knowledge at the lexical level, ensuring that the participants knew the critical nouns and their grammatical gender in Greek.

### 5.1. Eye Tracking Registration

In the current study, we use the mixed effects model for the analysis of the data (Raudenbush and Bryk 2002), which has been implemented using R.

The objective was to predict the behavior of three dependent variables: first pass duration of fixations, selective regression-path duration of fixations, and total duration of fixations. These three represent three of the "big four" durational measures (Godfroid 2020, p. 248) prominent in the literature, thereby facilitating the comparison of the present study with previous research. First pass duration is the total duration of the fixations inside the area of interest during the first pass. It is a measure of early stages of processing understood to signify lexical access. Although it is considered less informative for L2 populations (Godfroid 2020, p. 220), it remains nodal in research, including work on grammar acquisition (Clahsen et al. 2013; Felser and Cunnings 2012). Selective regression-path duration is the total duration of the fixations inside the area of interest before any fixation occurs in any area of interest progressive to this one. Notably, this duration includes fixations that might land on preceding parts of the text. As such, it is considered to represent integration of the area of interest (AOI), which could entail resolving ambiguity, ungrammaticality, and similar challenges. The measure holds an intermediary status between early and late measures (Conklin and Pellicer-Sánchez 2016), acting as a "hybrid measure" as described by Godfroid (2020, p. 223). It has been widely employed in research on grammar acquisition (e.g., Clahsen et al. 2013; Felser and Cunnings 2012; Lim and Christianson 2015) and is particularly relevant for the present study, as agreement violations are anticipated to introduce processing difficulties. Total duration of fixations is the sum of all fixations, even for multiple visits, inside the area of interest. It is a standard measure of late processing stages in text-based eye-tracking encompassing "global effects" (Godfroid 2020, p. 226) since it aggregates many of the early and late measures the researcher would register in a study. Consequently, any sensitivity to agreement violations evident in the first pass or the selective regression-path duration of fixations is expected to manifest in the total duration of fixations. By detecting an ungrammaticality effect across initial and late stages of processing using multiple measures, we gain confidence in the evidence, especially if the effect is observed throughout the course of the processing.

A logarithmic transformation was applied to the data due to its lack of a normal distribution. As the mixed effects model is robust with regard to outliers, no identification or modification of these was performed (Keating and Jegerski 2015). The parameters that are used as fixed effects are the grammatical (grammatical, ungrammatical) and phonological (phonological, non-phonological) status of the structure. The interaction between grammatical and phonological status has been examined as well. On the other hand, participants and items have been defined as random effects and are used as random intercepts, while no random slope has been utilized. Random slopes were used in conducting the analysis; however, it was concluded that the AIC/BIC parameters were smaller for the current

model and the *p*-values indicated that there was no statistical significance between them. Consequently, the simpler model has been chosen.

For each fixation time, two different cases have been analyzed, namely that of determiner–noun and adjective–noun, separately[10]. The results of the analysis are presented in the following tables.

### 5.1.1. Adjective–Noun Agreement

Table 5 shows the mean values of fixation times for both phonological and non-phonological conditions for adjective–noun. What can be extracted from this table is that for the ungrammatical errors there is an increase in the fixation times across phonological conditions which is observed in late measures (i.e., selective regression path[11] and total duration of fixations). In contrast, no sensitivity is registered for the early stage of processing agreement violations.

**Table 5.** Fixation times for adjective–noun.

| ADJ–N: Fixation Times Across Conditions | | | | | | |
|---|---|---|---|---|---|---|
| | | First Pass Duration (ms) | | Selective Regression Path Duration (ms) | | Total Duration of Fixations (ms) |
| | | Mean | StD | Mean | StD | Mean | StD |
| Phonological | Grammatical | 401.04 | 270.92 | 465.15 | 305.86 | 571.46 | 497.82 |
| | Ungrammatical | 380.36 | 249.09 | 494.57 | 349.47 | 708.03 | 538.51 |
| | Difference | −20.68 | | 29.42 | | 136.56 | |
| Non-phonological | Grammatical | 409.54 | 295.37 | 462.15 | 306.25 | 585.61 | 506.67 |
| | Ungrammatical | 411.43 | 298.94 | 552.95 | 392.82 | 778.65 | 626.87 |
| | Difference | 1.90 | | 90.80 | | 193.04 | |

By applying the mixed effects model, the results presented in Table 6 are acquired. The reference categories (intercept) consist of the duration of fixations on items containing grammatical and phonological agreement.

**Table 6.** Mixed effects model results for adjective–noun.

| Mixed-Effects Model (ADJ–N) | | | | | |
|---|---|---|---|---|---|
| | Predictor | Estimate (β) | Standard Error (SE) | t Value | *p* |
| First Pass Duration | Intercept | 5.791 | 0.074 | 78.218 | <0.001 |
| | Phonological | −0.013 | 0.076 | −0.177 | 0.860 |
| | Grammaticality | −0.068 | 0.081 | −0.842 | 0.401 |
| | Phono*Gram | 0.073 | 0.111 | 0.655 | 0.513 |
| Selective Regression Path Duration | Intercept | 5.951 | 0.082 | 72.923 | <0.001 |
| | Phonological | −0.031 | 0.071 | −0.445 | 0.657 |
| | Grammaticality | 0.011 | 0.075 | 0.140 | 0.889 |
| | Phono*Gram | 0.110 | 0.103 | 1.073 | 0.285 |
| Total duration of fixations | Intercept | 6.099 | 0.088 | 69.386 | <0.001 |
| | Phonological | 0.006 | 0.070 | 0.083 | 0.934 |
| | Grammaticality | 0.163 | 0.072 | 2.251 | 0.026 |
| | Phono*Gram | 0.059 | 0.102 | 0.582 | 0.561 |

The interaction between grammatical and phonological status did not have a statistically significant main effect on any of the examined fixations.

Regarding first pass duration and selective regression path duration, there is also no statistically significant main effect by neither phonological nor grammatical status. This indicates that fixation times are not statistically different for grammatical and ungrammatical sentences, as well for phonological and non-phonological ones.

In contrast, for the total duration of fixations, there is a statistically significant main effect of grammaticality. To be more specific, the logarithmic total duration of fixations regarding ungrammatical cases is 0.163 (*p* = 0.026), higher when compared to grammatical ones, meaning that the participants spent more time when processing violations of grammaticality irrespective of the phonological status of the area of interest (AOI). The significance of this result is also enhanced by the Bayes factor which has been calculated to be approximately 4.11. According to Lee and Wagenmakers' (2014) classification scheme it is moderate evidence that favors the current against the null model. It may also be concluded that Bayes factors significantly favor the null model regarding the phonological effect as well as the interaction between phonological and grammatical status. This indicates that indeed, there is no effect of any factor for first pass and regression duration, while for the total duration only grammaticality has been identified to have an effect. Table 7 shows the Bayes factors for all the effects.

**Table 7.** Bayes factors for Adj–N effects.

|  | Adj–N | | |
|---|---|---|---|
|  | **Interaction** | **Grammaticality** | **Phonological** |
| First Pass | 0.0000004 | 0.0056446 | 0.0053113 |
| Regression | 0.0000010 | 0.0119994 | 0.0050999 |
| Total Duration | 0.0002467 | 4.1067960 | 0.0054480 |

The overall participants' processing routines is displayed in Figure 1.

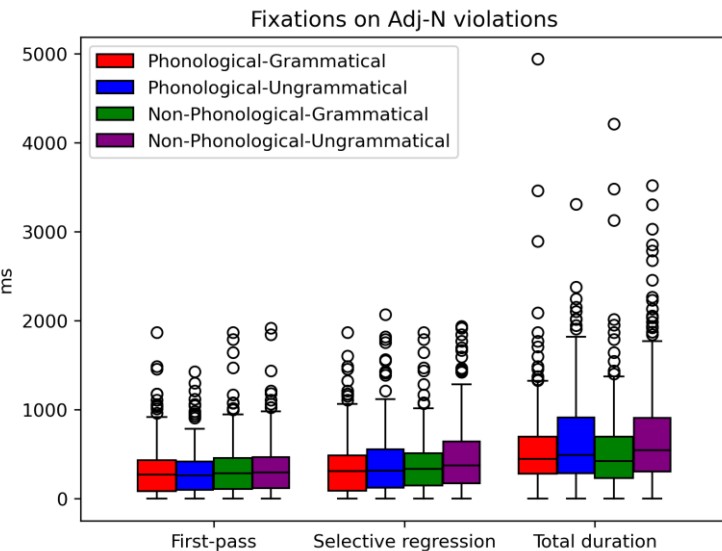

**Figure 1.** Fixations on Adj–N violations.

5.1.2. Determiner–Noun

Table 8 shows the mean values of fixation times for both phonological and non-phonological conditions for determiner–noun agreement. Similar to Adj–N dependencies, fixation times are longer for ungrammatical conditions, regardless of the of the sentences' phonological status. More specifically, the effect of the ungrammaticality becomes evident

in late measures (i.e., selective regression path and total duration of fixations). Conversely, participants do not demonstrate sensitivity to agreement violations during the early stages of processing.

**Table 8.** Fixation times for determiner–noun.

| | | DET–N: Fixation Times across Conditions | | | | | |
|---|---|---|---|---|---|---|---|
| | | **First Pass Duration (ms)** | | **Selective Regression Path Duration (ms)** | | **Total Duration of Fixations (ms)** | |
| | | **Mean** | **StD** | **Mean** | **StD** | **Mean** | **StD** |
| Phonological | Grammatical | 321.78 | 222.04 | 356.98 | 227.46 | 431.01 | 315.72 |
| | Ungrammatical | 362.22 | 249.52 | 493.19 | 419.90 | 645.97 | 549.62 |
| | Difference | 40.44 | | 136.21 | | 214.96 | |
| Non-phonological | Grammatical | 364.05 | 306.98 | 420.00 | 332.41 | 468.39 | 389.34 |
| | Ungrammatical | 381.13 | 311.21 | 499.24 | 399.44 | 646.76 | 538.23 |
| | Difference | 17.08 | | 79.24 | | 178.37 | |

Table 9 presents the findings of the mixed effects model for the Det–N condition. Similar to the Adj–N, the reference categories are phonological and grammatical.

**Table 9.** Mixed effects model results for determiner–noun.

| | Mixed-Effects Model (DET–N) | | | | |
|---|---|---|---|---|---|
| | **Predictor** | **Estimate (β)** | **Standard Error (SE)** | **t Value** | ***p*** |
| First Pass Duration | Intercept | 5.543 | 0.075 | 74.018 | <0.001 |
| | Phonological | 0.067 | 0.075 | 0.890 | 0.374 |
| | Grammaticality | 0.107 | 0.083 | 1.293 | 0.197 |
| | Phono*Gram | −0.056 | 0.112 | −0.499 | 0.618 |
| Selective regression path duration | Intercept | 5.635 | 0.084 | 66.826 | <0.001 |
| | Phonological | 0.092 | 0.076 | 1.220 | 0.223 |
| | Grammaticality | 0.264 | 0.085 | 3.124 | 0.002 |
| | Phono*Gram | −0.076 | 0.114 | −0.665 | 0.506 |
| Total duration of fixations | Intercept | 5.767 | 0.090 | 64.304 | <0.001 |
| | Phonological | 0.084 | 0.074 | 1.133 | 0.258 |
| | Grammaticality | 0.310 | 0.083 | 3.747 | <0.001 |
| | Phono*Gram | −0.047 | 0.113 | −0.416 | 0.677 |

As seen in Table 9, there is no interaction between the phonological and grammatical status that would have a statistically significant effect.

Regarding first pass duration, there is no statistically significant main effect by either phonological or grammatical violations.

However, for selective regression path duration and the total duration of fixations grammaticality shows a statistically significant main effect. More specifically, for selective regression, the logarithmic value of fixations for ungrammatical cases is 0.264 ($p$ = 0.002) higher than it is for grammatical cases. As shown in Figure 2, this difference is observed for both phonological and non-phonological concords, indicating that the phonological status does not significantly impact the processing of the AOIs.

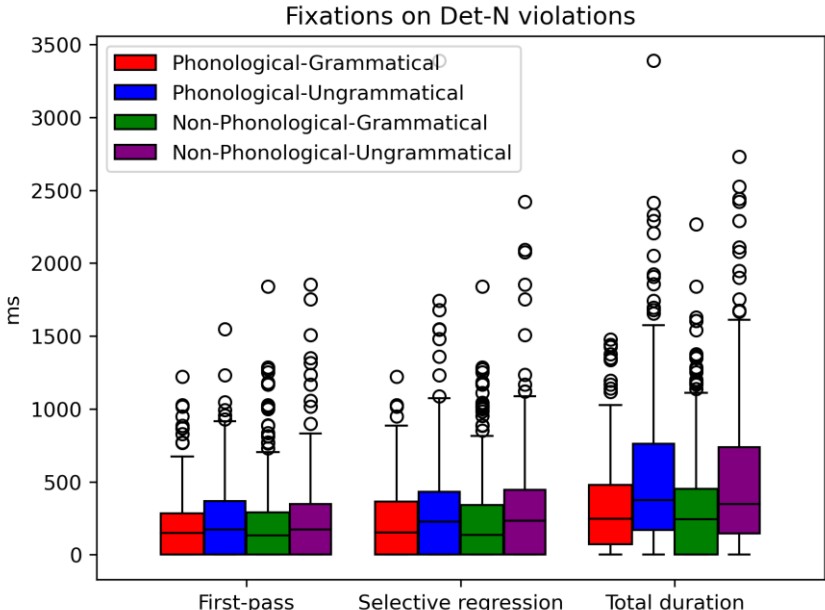

**Figure 2.** Fixations on Det–N violations.

Similarly, for the total duration of fixations, the logarithmic value of fixations for un-grammaticality is 0.310 ($p < 0.001$) higher than it is for grammatical cases, a value which is not affected by the phonological status of the violation. In a similar manner as Adj–N, the Bayes factors have been calculated, which for the statistically significant estimations of the mixed effects model indicate that there is moderate and extreme evidence that favor the current against the null-model. On the contrary, the very low Bayes factor values point out the negligible effect of the respective factors. Table 10 shows the Bayes factors for all the effects.

**Table 10.** Bayes factors for Det–N effects.

|  | **Det–N** | | |
| --- | --- | --- | --- |
|  | **Interaction** | **Grammaticality** | **Phonological** |
| First Pass | 0.0000015 | 0.0145252 | 0.0077943 |
| Regression | 0.0001336 | 5.4161020 | 0.0103457 |
| Total Duration | 0.0437027 | 426.0828000 | 0.0097375 |

As observed in Figure 2, the total duration of fixations indicate that the violation of grammaticality triggered the participants' attention. Interestingly, the phonological status of the Det–N dependencies did not have any impact on the participants' fixations. In Section 6 we elaborate more on these results.

*5.2. Post-Reading Questionnaire*

The post-reading questionnaire provided valuable feedback concerning the impact of the specific experimental design on the type of knowledge that was triggered during the reading-based study. This was a main concern which largely affected key decisions regarding the design of the critical stimuli. Notably, this led to the decision to integrate them within reading passages rather than using standalone one-sentence stimuli. Table 11 displays the responses to each question. Recall that this tool was adapted from Spino's (2022) design to which these responses are contrasted in the discussion.

**Table 11.** Participants' responses to the post-reading questionnaire.

| QUESTIONS | |
|---|---|
| ***Q1.****Did you notice anything strange about the short texts you read? If so, what?* | Participants 24 |
| Percent reporting any grammatical violations (including but not limited to gender)<br>Percent reporting gender violations<br>Percent reporting Det–N violations<br>Percent reporting Adj–N violations | 29%<br>13%<br>0%<br>0% |
| ***Q2.****Were there any grammatical errors in the short texts you read? Please check:*<br>*Yes No* | |
| Percent reporting noticing errors | 33% |
| ***Q3.****What types of grammatical errors did you notice? Please list all the errors you remember and provide examples when possible.* | |
| Percent mentioning gender violations specifically<br>Percent specifying Det–N violations<br>Percent specifying Adj–N violations | 13%<br>9%<br>0% |
| ***Q4.****Please check off the types of errors you noticed in the texts. If you are unsure as to what something is, please ask the researcher.* | |
| Percent who checked off noticing<br>*Determiner–noun agreement violations*<br>*Spelling mistakes*<br>*Subject–verb agreement violations*<br>*Illicit use of some nouns*<br>*Adjective–noun agreement violations*<br>*Tense–adverb violations*<br>*Illicit use of fonts* | <br>29%<br>4%<br>4%<br>0%<br>9%<br>0%<br>0% |
| ***Q5.****What is the percentage of the errors in the passages you read?* | |
| Average percent of errors in the passages | 21% |

Participants who answered all the questions were those who had checked off the 'Yes' option in Q2. There were eight (33%) participants who noticed grammatical errors. Only three of them answered that they processed grammatical violations in Q3 which did not provide the error types. Two participants (9%) could recall the type of violations. When error typology was provided by Q4, seven participants (29%) identified Det–N violations and one participant indicated both structures (Det–N and Adj–N) were violated. For Q5, participants estimated the density of the errors in the passages they had read. The eight participants who answered Q5 slightly underestimated the density of errors by indicating a proportion of 21%.

*5.3. Vocabulary and Gender Assignment Post-Test*

The vocabulary test is the last that was administered in the study. With this test the presence of the gender feature at the lexical level was examined. As expected, the participants were familiar with the nouns used in the study: they assigned the nouns to the correct gender value with an accuracy of 19.24 out of twenty critical items. The task posed a challenge for three participants whose scores on the task of placing the correct article fell below 85%. The respective data (i.e., the nouns that did not receive the correct article) were not included in the analysis of the eye-tracking registration. The accuracy rate was even higher in the translation of the nouns with the participants achieving an accuracy rate of 19.88 of twenty nouns.

The aim of having a reliable measure to assess gender assignment is twofold: firstly, it facilitates the study of the Det–N structure in terms of syntactic computations, shedding light on potential L1 interference in how L2 learners process the article. Secondly, it could

inform error taxonomy (see Tantos and Amvrazis 2022) in the context of corpus annotation since higher sensitivity to Det–N violations (e.g., in the pair[12] *ti*$_{FEM}$ *δema*$_{NEUT}$, *mikri*$_{FEM}$ *δema*$_{NEUT}$), could suggest a classification of the anomaly as an agreement error, rather than an assignment one.

## 6. Discussion

This study aimed to investigate the use of grammatical gender in Russian-speaking learners of Greek during online reading comprehension. A twofold approach was employed. Firstly, we examined whether the grammatical and phonological status could predict sensitivity to grammatical violations in Det–N and Adj–N agreement dependencies during online processing, as measured by eye-tracking technology. Secondly, we explored through triangulation the association of the L2ers' online eye gaze behavior with the production of gender marking on determiners and adjectives by participants with the same profile. Besides from eye tracking measures, the study design also utilized expressive measures as well. The use of an offline vocabulary task tested the L2ers' morphological knowledge upon the features encoding grammatical gender at the lexical level. The results of the vocabulary and gender assignment task indicate that Russians' developing grammar of Greek at the intermediate level are aware of the gender feature at the lexical level and its respective morphological marking since the subjects performed at ceiling in this task[13]. Given the requirement of target-like performance in this task it was deemed safe to explore the computations operating at the syntactic level through eye-tracking experimentation. Note that the participants' L1 employs agreement computations but only in the Adj–N structures, thus allowing for the inquisition of the role of the L1 in the course of acquisition.

The eye-tracking registration was designed to circumvent explicit activation by incorporating agreement violations within a reading passage, rather than presenting them in separate sentences. The post-reading questionnaire data indicates that this design successfully suppressed awareness of the critical stimuli. In contrast to the results reported by Spino (2022), only 33% of the participants in the current study recalled the violations when asked to check them off in a list. In the same question, 64% of the participants in Spino's (2022) study reported awareness of the violations. The current study's design of the critical stimuli, as well as the use of grammatical non-critical distractors[14], likely accounts for the difference in results.

Despite the minimized activation of explicit awareness, the eye-tracking reading paradigm showed that agreement violations had been noticed by the L2ers in both Det–N and Adj–N structures. Critically, though, the violation was detected earlier in the Det–N condition, with longer fixations being recorded by both the selective regression path and the total duration measures. More precisely, the L2ers had noticed the violation after 496 ms (average) in the Det–N structure and this awareness was progressively enhanced when fixations had reached 646 ms. In cognitive terms, the violation was detected upon the word integration, a function that is accomplished before the eye movements land to an element forward to the AOI (i.e., selective regression path measure) (Clifton et al. 2007). On the other hand, violations in the Adj–N structure required an average of 743 ms of fixation to be noticed, a measure that corresponds to the total duration of all fixations which also includes regressive fixations on the AOI. An indirect but accurate contrast between the two structures comes from the estimate, β, an indicator that measures the difference between the grammatical and the non-grammatical stimuli. The estimate in the Det–N (β = 0.310) condition was found to be twice as high as the respective estimate for the Adj–N (β = 0.163) condition. The above observation corroborates the reports in the post-reading questionnaire where the Det–N violations were by far the most salient type of the violations that could be recalled by the learners. Moreover, this data validates the corpus findings in which the Adj–N dependencies evoked about twice as many errors as were produced in the Det–N condition. Within the Feature Reassembly Hypothesis (Lardiere 2009) it is anticipated that the L1 settings could be transferred in the L2, as evinced in both the corpus and the current study by the general performance of the L2 learners. The abstract gender

feature is present in the grammar of intermediate Russian learners of Greek. Moreover, the FRH presumes that the L1 setting could be neutralized as a result of the reassembly process in which the learner may redistribute even the feature values that are shared by the L1 and the L2. It follows that, even though Russian and Greek employ agreement computations in the case of the Adj–N dependency, this does not guarantee its morphological realization in the way it is expressed in Greek. A plausible explanation comes from the fact that the features that manifest agreement are not always detectable in Greek, since the features of gender, number and case are extensively conflated, thus not meeting the requirement of a formal contrast that would render the features salient[15] (Lardiere 2009; see also Ayoun and Maranzana 2022). At the same time, the non-transparent nature of the morphological marking of gender in Russian could further reduce the attention of Russian learners on the respective markers that establish agreement in Greek (see Hopp and Lemmerth 2018 for similar effects on predictive reading). The challenge for the learner is further escalated by the well-known assumption that agreement dependencies hold limited communicative value[16], especially for non-advanced learners, and as such the respective cues do not receive attention during processing (VanPatten 1996, 2004). The ambiguities of the morphemes that are used in agreement computations and their lack of communicative effect complement the FRH by expanding its scope from production to real-time comprehension[17] and by shedding light on the causes of the weakened sensitivity to syntactic dependencies which are common to both L1 and L2 (i.e., Adj–N). On the other hand, given (a) the learning approach in the instructed environment where the nouns are lemmatized by the teacher and the course books contain the appropriate article, and (b) the robust input of the Det–N concord outside the classroom, we expect that Russian learners of Greek can compensate for the lack of determiners in their L1. It is reasonable that the associations between determiners and nouns become stronger in the L2ers' mental lexicon and more available in real-time processing. In the above context, the differences observed concerning Det–N and Adj–N structures receive support in formal and pedagogical terms.

The results reported by this study are partly in line with those reported by Spino (2022), in which participants were sensitive to violations of both structures but only the Det–N domain reached statistical significance. Keating (2009) has also found sensitivity to agreement violations involving Adj–N concordance within the DP, the type of dependency we investigate in this study. However, Keating does not include contrasts between Adj–N and Det–N structures. Beyond these studies, which fall under the same experimental paradigm, L2 learners have shown higher accuracy in Det–N agreement in both production (Franceschina 2001; De Garavito and White 2002; White et al. 2004) and comprehension tasks (Foucart and Frenck-Mestre 2011; Dowens et al. 2011). Although there are no comparable contrasts (i.e., Det–N vs. Adj–N) in the studies that focus on the acquisition of grammatical gender in Greek, the general pattern is congruent with the results the present study delivers. While the Det–N domain seems to be ultimately achieved with native-like patterns displayed in some cases (e.g., Chondrogianni 2008), mastery of Adj–N is acquired over time, with patterns of overgeneralization and phonological agreement being especially problematic (e.g., Agathopoulou et al. 2008; Tsimpli et al. 2007). The present work also explored the latter, but it did not find significant effects on the processing routines followed by L2 learners. Unlike the findings of the corpus study and previous work with offline measures (Agathopoulou et al. 2008), the eye-tracking registration has shown that sensitivity to gender violations is equally affected by both phonologically matched and mismatched concordances. We assume that the explicit knowledge exploited by an intermediate learner during offline tasks (e.g., written production, cloze test) can lead to conscious constructions in which the repetition of the noun suffix onto the agreeing adjective assimilates a pattern observed in the input received in or outside the classroom.

The main limitation of the present study is inherent in its priority to triangulate the results of a previous corpus study. In order to meet the requirements of a comparable design that would allow for meaningful contrasts, the use of short reading passages was a prerequisite. Although the length of the critical items and the endings were balanced, the

sample is not adequate to support an analysis on a morpheme-basis. A future work on the role of the particular morphemes participating in the agreement computations should control this factor and set a design to test certain variables (e.g., overgeneralized morphemes, transparency, distributional bias across gender, number, and case). The above passages, however, not only assimilated the learners' productions in the corpus but also increased the ecological validity of the data elicitation in the eye-tracking registration. It seems that the use of text-based, instead of sentence-based, stimuli can impact to what extent explicit knowledge is triggered. Comparing the results of the post-reading questionnaire with those reported by Spino (2022), it turns out that the stimuli presentation is a key factor, at least when sensitivity to grammatical violations is being measured. Furthermore, utilizing a comprehension-based secondary task, as opposed to a metalinguistic one, appears to minimize activation of explicit knowledge, particularly following text-based processing (see also discussion in Leeser et al. 2011).

Finally, one pedagogical implication derived from this study's design is that a variety of tasks should be used to elicit grammatical violations. The use of context, whether in a written, oral production or a reading task, is more capable of bringing the implicit knowledge to light. By inducing rule-based knowledge using focus-on-forms tasks (e.g., cloze tests, grammaticality judgments) we may get a distorted view of the learner's interlanguage.

**Author Contributions:** Conceptualization, N.A. and A.T.; methodology, A.T., N.A., K.A. and K.K.; software, K.A.; validation, K.A., N.A. and A.T.; formal analysis, K.A. and A.T.; investigation, A.T. and N.A.; resources, A.T. and N.A.; data curation, N.A. and A.T.; writing—original draft preparation, N.A., A.T., K.A. and K.K.; writing—review and editing, N.A., K.A. and A.T.; visualization, K.A.; supervision, A.T.; project administration, A.T.; funding acquisition, A.T. All authors have read and agreed to the published version of the manuscript.

**Funding:** The research work was supported by the Hellenic Foundation for Research and Innovation (H.F.R.I.) (https://www.elidek.gr/en/call/6489/, accessed on 28 October 2023) under the "First Call for H.F.R.I. Research Projects to support Faculty members and Researchers and the procurement of high-cost research equipment grant" (Project Number: 3161).

**Institutional Review Board Statement:** The study was conducted in accordance with the Declaration of Helsinki, and approved by the Institutional Review Board (or Ethics Committee) of the Aristotle University of Thessaloniki (protocol code 62503/2022 and date of approval: 09/03/2022).

**Informed Consent Statement:** Informed consent was obtained from all subjects involved in the study.

**Data Availability Statement:** Not applicable.

**Conflicts of Interest:** The authors declare no conflict of interest.

## Appendix A. Example of a Text Followed by Comprehension Task (2 Consecutive Screens)

Γεια σου Άνια! Είμαι στην Αθήνα αυτές τις μέρες. Ωραία πόλη, όμως πάλι έχω πρόβλημα με το *ζέστη* εδώ. Το κλίμα στη *χώρα* μου *είναι* πολύ διαφορετικό. Επιτέλους, *αύριο θα πάω* στη θάλασσα, αλλά η *θέμα* εκεί είναι τα λεωφορεία. Είναι *πάντα γεμάτα!* Χτες *πήγα* στο *αγορά* για μαγιό αλλά *δεν βρήκα* το χρώμα που *ήθελα.* Θυμάσαι τον Χουάν; Κάναμε παρέα τις *μήνες* που μέναμε στην εστία. Θα βγούμε το βράδυ για ένα ποτό και *ίσως πάμε* στον *μέρος* που βγαίναμε όλοι μαζί.
Φιλιά, Φρίντα!

Hello Ania! I am in Athens these days. It is a beautiful city, but I'm having trouble with the heat here again. The climate in my country is very different. Finally, tomorrow I will go to the beach, but the problem there is the buses. They are always full! Yesterday, I went to the market for a swimsuit but I didn't find the color I wanted. Do you remember Juan? We hung out during the months we stayed in the dormitory. We will go out tonight for a drink and maybe go to the place where we all used to hang out together.
Kisses, Frida!

Comprehension task

Πού θα πάνε ίσως η Φρίντα με τον Χουάν απόψε;

Α. Στην παραλία.

Β. Στην αγορά.

Γ. Στο μέρος που συνήθιζαν να βγαίνουν μαζί.

Where will Frida and Juan possibly go tonight?

(a) To the beach.

(b) To the market.

(c) To the place where they used to hang out together.

### Appendix B. Example with AOIs in the Text-Based Paradigm

Γεια σου Άνια! Είμαι στην Αθήνα αυτές τις μέρες. Ωραία πόλη, όμως πάλι έχω πρόβλημα με το ζέστη εδώ. Το κλίμα στη χώρα μου είναι πολύ διαφορετικό. Επιτέλους, αύριο θα πάω στη θάλασσα, αλλά (η) (θέμα) (εκεί) είναι τα λεωφορεία. Είναι πάντα γεμάτα! Χτες πήγα στο αγορά για μαγιό αλλά δεν βρήκα (το) (χρώμα) (που) ήθελα. Θυμάσαι τον Χουάν; Κάναμε παρέα τις μήνες που μέναμε (στην) (εστία). Θα βγούμε το βράδυ για ένα ποτό και ίσως πάμε στον μέρος που βγαίναμε όλοι μαζί.

Φιλιά, Φρίντα!

| | | | |
|---|---|---|---|
| Det-N critical item: | (η)$_{AOI1}$ | (θέμα)$_{AOI2}$ | (εκεί)$_{(AOIspill over)}$ |
| | (the) | (problem) | (there)... |
| Det–N control item: | (το)$_{AOI1}$ | (χρώμα)$_{AOI2}$ | (που)$_{(AOIspill over)}$ |
| | (the) | (color) | (that) |
| Det–N filler: | (στην) | (εστία) | |
| | (in the) | (dormitory) | |

## Notes

[1] Throughout the paper, the term 'determiners' is conventionally used to refer to articles. Although Russian employs determiners (e.g., demonstratives) that agree with the noun in gender, case, and number, as pointed out by a reviewer, it does not have articles.

[2] The presence of an adjective in agreement with a noun is not always a reliable marker of gender because of syncretism across gender, number and case. The examples in (3) are indicative, illustrating only one declension in nominative case of the singular number.

[3] Besides the placement test, the teachers could indicate the participants that were aligned in terms of their proficiency level on the basis of both the daily contact and the assessment tools utilized in the instructed context.

[4] The GLCII is freely available at https://glc.lit.auth.gr/app/GLC_Gateway (accessed on 28 October 2023)

[5] The suffixes that are responsible for 78% of the 208 agreement errors in the corpus are:

-es (masculine or feminine in plural),
-ma (neuter in singular, but it conflates as –(m)a with the following category),
-a (feminine singular, neuter plural, masculine singular in accusative),
-i (feminine singular, masculine plural, neuter singular); the phonological value is the same although expressed through different characters,
-os (masculine, feminine, neuter in singular).
Therefore, the items are not standardized for gender but for suffix.

[6] Morphemes and syntactic position were standardized. However, these were not taken as factors in the analysis of the study due to design restrictions (i.e., size of the experiment). A future increase in the participants sample could enable their exploration.

[7] Of the 80 nouns, 20 control nouns matched the 20 critical items in terms of morphemes and length. The remaining 40 nouns appeared in grammatical contexts, either with a determiner or an adjective.

[8] Data from texts with inaccurate responses in the comprehension task were not excluded from the analysis. This is because the task was not designed to validate the critical items, but rather to enhance the ecological validity of the text-based experiment and ensure that the participant was focused on the process.

[9] The form is available at: https://docs.google.com/forms/d/e/1FAIpQLSeFnnA-0JpYG9zHvy-QpJi8wqBp-pIF4uQb6sbfVcj7YGhWVA/viewform (accessed on 28 October 2023)

[10] One of the reviewers suggested that running a single test (lmer) might enable a statistical comparison between the two structures. However, the Det–N and Adj–N concords aren't directly comparable, especially in an online context. In Greek, adjectives differ

in length from articles. For instance, an adjective–noun structure has a minimum of 2 + 2 syllables, as seen in 'fti-no spi-ti' ("cheap house"), whereas the corresponding article–noun structure consists of 1 + 2 syllables, as in 'to spi-ti' ("the house"). Given this, it is expected that adjectives will produce longer fixations, not just because of their length but also due to their semantic content. This is why we implement comparisons between fixation measures exclusively within the same structure. Although indirect, an accurate contrast between the two structures (Adj–N vs. Det–N) can be gleaned from the estimate (β). This indicator measures the difference between grammatical and non-grammatical stimuli (see Discussion).

11  Selective regression-path duration is considered a late measure by a number of studies (see discussion in Godfroid 2020).

12  In the literature it is hard to tease apart gender assignment from gender agreement. The issue is of vital importance in the context of corpus annotation where the researcher has to classify errors like the one in the example. Having such insights from eye-tracking in conjunction to the offline vocabulary test gives the annotator more confidence when annotating grammatical gender in GLCII. A reviewer commented that the results of such a measure cannot conclusively determine the nature of the violation. We concur with this comment. Informing corpus annotation was a byproduct of the present study, with our intention being to provide evidence for plausible interpretations of error ambiguity.

13  However, activation of explicit knowledge as a task effect could not be excluded in the case of a fill-in-the-blanks vocabulary test.

14  Half of the noncritical stimuli were ungrammatical in the Spino's design.

15  Lardiere (2009) does not make any specific claim regarding the acquisition of particular φ-features (see also Ayoun and Maranzana 2022, p. 118). Nevertheless, the lack of specificity in this regard does not affect the interpretation of the results in the context of this study.

16  The current study examines inanimate nouns. Given that their grammatical gender is arbitrary, the agreement is rendered a formal function without impact at the semantic level. In this context the agreement is of low value in terms of communicative efficacy.

17  The choice of the specific account is based on the broader design of this study whose critical items were derived from written (and, in the near future, spoken) productions. Even though the Shallow Parsing Hypothesis (Clahsen and Felser 2006) seems to better fit our findings, we opted for a unified approach that could account for the expressive and receptive measures employed.

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
