# Peer review of "Exploring Grammatical Gender Agreement in Russian Learners of Greek: An Eye-Tracking Study"

_languages, doi:10.3390/languages8040265_

Round 1
Reviewer 1 Report
Comments and Suggestions for Authors
This study examines if Russian L1-Greek L2 learners are sensitive to gender incongruencies in determiner-noun and adjective noun pairings using an eyetracking paradigm while participants read passages for comprehension. Both Greek and Russian feature grammatical gender, but they differ in how gender is marked, and therefore implications about how L1 processing can affect L2 parsing are discussed. Nevertheless, this study is innovative in that it employs a Greek L2 learner corpus to motivate the design of their stimuli, uses contextualized short passages, and triangulates the online data with an offline post-reading questionnaire to explore if longer fixations on agreement violations could be associated with explicit awareness of the study’s manipulation. Results revealed that Russian L1-Greek L2 learners are sensitive to adjective-noun gender manipulations in the late measure of total duration time, and to noun-adjective gender manipulations in both late measures (total duration and selective regression path). This sensitivity was not affected by phonological conditions. Results from the post-reading questionnaire showed that only 33% of participants reported noticing grammar mistakes and only 9% of them was able to recall the type of error. When the type of error was provided, only seven participants indicated determiner-noun errors and only one marked adjective-noun errors. Overall, these findings suggest that intermediate Russian learners of Greek are aware of the gender feature and its respective morphological computations, and that the study design (with contextualized passages and post-reading questionnaire) suppressed awareness of the critical stimuli.
General comments:
The manuscript is very well-written and easy to follow. I commend the authors for spending so much time and effort designing the stimuli, which I believe increases the ecological validity of the data obtained. I do not see any major concerns with the study design, or theoretical argumentation, but I have a few suggestions to improve the manuscript:
Abstract
This sentence (lines 19-21), particularly the underlined section is hard to parse: “…the gender agreement data suggests that the abstract gender feature is present in the developing grammar of Russian learners of Greek, while the agreement computations are operating for the same participants of the study.” Do you mean that learners possess both: (a) knowledge of lexical gender, and how it should agree with nouns and adjectives? Please rephrase to ensure clarity.
Introduction
Line 78: Please spell out GLCII for the first time. It is spelled out once before, but in the abstract.
Syntactic background: gender
Line 219: Does Ds stand for determiners? Please spell out.
Study
Participants
According to more contemporary terminology, it is not accurate to say that the Russian speakers were monolingual (line 327), I recommend the use of the term “monolingually raised Russian speakers”, since these participants are, to some extent, proficient in Greek.
Preliminary considerations on the design
Similar to the current study, Beltrán-Lopez & Dussias (2023) in Bilingual Approaches to Bilingualism (forthcoming) used a corpus to create their eyetracking stimuli. If I recall correctly, they used sentences verbatim, but I believe this study is worth citing as it sets a precedent together with the present study for future studies to adopt such an approach when creating critical stimuli in sentence processing research.
https://www.researchgate.net/publication/372336249_Heritage_speakers%27_processing_of_the _Spanish_subjunctive_A_pupillometric_study?fbclid=IwAR2En8vu5Re5Io9fyMIm3- iVmKbBEmm8cgUpIK6ObVNkj-JZ_8T-7qTvQd8
Because of the unconventional stimuli design approach, several aspects of the stimuli are not counterbalanced as traditionally expected (e.g., nouns not being equally dispersed by formal aspects such as gender). I wonder if the authors could control for noun frequency, since frequency has been found to affect language processing, particularly with L2 learners. Hopp (2014, 2018) examined factors at the world level and how they can have implications for processing at the phrase and sentence level. Frequency is one of them. If there is a Greek corpus out there with frequency data, it may be a good idea to examine the frequency of the nouns used in the study to determine if there is a significant difference between those used with determiners and adjectives. If there is a difference, the authors can always add frequency as a covariate in the model (as in Fernández Cuenca & Jegerski, 2022).
Fernández Cuenca, S., & Jegerski, J. (2022). A role for verb regularity in the L2 processing of the Spanish subjunctive mood: Evidence from eye-tracking. Studies in Second Language Acquisition, 1-30. doi:10.1017/S027226312200016X
Hopp, H. (2014). Working memory effects on the L2 processing of ambiguous relative clauses. Language
Acquisition, 21, 250–278.
Hopp. H. (2018). The bilingual mental lexicon in L2 sentence processing. Second Language, 17, 5–27.
Materials and methods
Typo in line 370 (e.ye tracking)
Could you elaborate on what “attribute settings” mean in line 397?
In lines 401-402, the authors say “The ratio of critical to non-critical items was 25%.” What did the non-critical items look like? Were they distractors, fillers? It is hard to decipher given the innovative stimuli design. It would be important to clarify this by providing a clear description of how this was done for future researchers to use this study as a reference point.
For those of us unfamiliar with Greek and Russian, it would be helpful to have English translations or linguistic annotations for table 4 (after line 411). This table is fundamental to understand the study’s conditions, but the way it is set up right now, forces the reader to go back to table 1 to remember which suffixes correspond to which grammatical gender.
If I understood correctly, there is a comprehension-based question after each passage. Could you provide an example of a passage and a comprehension question to illustrate a critical item and to show that the comprehension questions did not target the critical stimuli directly?
In line 418, the authors mention that a minimum of 80% accuracy in responses was established to determine if participants had properly read the texts (passages). Were the passages with an inaccurate response included as part of the final data pool that was analyzed? Please specify, if they were discarded indicate and what percentage of the total data pool it was.
Despite the post-reading questionnaire being a great method to triangulate level of awareness and validity of the online data collected, the terminology used (e.g., illicit use of certain nouns or tense-adverb violations) seems to be very technical. Did you check to see if participants were familiar with this terminology? Were they linguistics students? This could be a limitation explaining why only 3 of the participants (out of the 8 that reported seeing errors) were able to identify which particular kind of error they were.
Results
Eyetracking registration
It would be very helpful to see the distribution of words in a critical region of interest. Was each word a region of interest? Were the determiner and noun (for example) two adjacent areas of interest? Was there no analysis of a spill-over region?
Line 529-530, please elaborate on why these types of measures were chosen for the particular target constructions examined in the study. For example, total dwell time is a good late measure to examine sensitivity to higher-level factors related to syntax, semantics, and pragmatics, whereas first fixation is often used to examine lexical factors such as word frequency, polysemy, etc. Clifton et al (2007) and Godfroid’s eyetracking book (2020) are good references for this.
Clifton, C., Staub, A., & Rayner, K. (2007). Eye movements in reading words and sentences. In R. P. G. Van Gompel, M. H. Fischer, W. S. Murray, & R. L. Hill (Eds.), Eye-movements: A window on mind and brain (pp. 241–372). Elsevier Science.
Godfroid, A. (2020). Eye tracking in second language acquisition and bilingualism: A research synthesis and methodological guide. Routledge.
What software was used to run the statistical analysis? R, SPSS? This information is relevant for how the most effective mixed effects models are built.
In lines 544-545, when discussing the adj-noun results, the authors state that there “ is an increase in the fixation times for both phonological and non-phonological conditions”. I encourage the authors to rephrase this and be more specific by stating that the increase could be
observed only with late measures (selective RP, and total duration fixation), but so much with the early measure of first pass (only in the right direction in the non-phonological condition). I would also use the same terminology for the det-noun results, in lines 570-571, where the difference in the expected direction (ungrammatical higher than grammatical) is present with both early and late measures. The timing of the sensitivity is relevant, since these results suggest that Russian learners of Greek were sensitivity to gender incongruencies faster (first pass) with noun-determiners than noun-adjectives pairings.
It is not clear why pairwise comparisons were run for what I believe is grammaticality and phonological conditions in lines 588-589, since there was not significant interaction (p=0.677).
Vocabulary and gender assignment post-test
In line 621, the authors say “the respective data were not included in the analyses”, is the respective data the vocabulary accuracy scores? And is “the analyses” referring to the eyetracking analyses? Meaning that they did not discard nouns that were familiars for specific participants? Please specify.
Results
In line 640, “the results indicate…” I would specify which tasks these results come from. Are these results from the post-reading questionnaire or the vocabulary and gender assignment task? Similarly, should “using an eyetracking paradigm” be added after “to explore the computations operating at the syntactic level” to clarify the task these results refer to?
In line 659, the authors state that “the violation was detected earlier in the Det-N condition”. My suggestion for establishing a difference between early and late eyetracking measures when discussing the descriptive stats (in lines 544-545 and 570-571) would connect very well with this observation. Also, in line 663, the authors mention “word integration” making reference to a particular eye movement measure (an early measure, in fact). It is for this reason that I suggest a more elaborated description in section 5.1 explaining why these three eyetracking measures were chosen for this study.
One of the most interesting points mention in the discussion can be found in lines 674-675, where the researchers point out that production learner data from the corpus matches the processing data found in the study. Perhaps the authors could elaborate more on this. For example, does the fact that Russian learners of Greek have more difficulty computing adj-noun incongruencies signify that this is a processing issue (at the input-intake stage) that affects the acquisition of how this feature works in Greek ? How does the post-reading questionnaire inform this proposal if only one participant was able to identify adj-noun errors? How does this relate to established theories in the field such as shallow processing or the feature reassembly hypothesis?
There is a no mention about proficiency level affecting sensitivity to gender agreement violations in the discussion, but this is briefly mentioned in the literature review. Was the proficiency level of the Russian learners of Greek comparable to that of those participants in previous studies?
Finally, the authors make a strong point towards the end of the discussion highlighting the importance of contextualizing the stimuli to increase the ecological validity of the data
elicitation. Although not exactly the same, there are a couple of studies that examined how the distractor task used to examine language processing can alter how much attention participants pay to the stimuli. I leave the references here in case the authors see the value in mentioning these studies to strengthen their argument:
Leeser, M., Brandl, A., & Weissglass, C. (2011). Task effects in second language sentence processing research. In P. Trofimovich & K. McDonough (Eds.), Applying priming methods to L2 learning, teaching, and research: Insights from psycholinguistics (pp. 179- 198). John Benjamins.
Tokowicz, N., & MacWhinney, B. (2005). Implicit and explicit measures of sensitivity to violations in second language grammar: An event-related potential investigation. Studies in second language acquisition, 27(2), 173-204
Author Response
Dear Reviewer 1.
We would like to express our gratitude for the time and effort you dedicated to reviewing our manuscript. Your insights and constructive suggestions have been instrumental in enhancing the quality of the paper. We have taken each of your comments into account and have made the necessary revisions to address them. Below, we elaborate on some of the comments you made for which we feel clarification is warranted.
Regards.
Alex Tantos, Nikos Amvrazis & Konstantinos Angelou

Reviewer 2 Report
Comments and Suggestions for Authors
See attached file.

There are several places where the English needs work. I am not opposed to having features of speaker dialect or L2 features in an article, but in many places this article is difficult to read and understand.
Author Response
Dear Reviewer 2.
We would like to express our gratitude for the time and effort you dedicated to reviewing our manuscript. Your insights and constructive suggestions have been instrumental in enhancing the quality of the paper. We have taken each of your comments into account and have made the necessary revisions to address them. Below, we elaborate on some of the comments you made for which we feel clarification is warranted.
Regards.
Alex Tantos, Nikos Amvrazis & Konstantinos Angelou

Round 2
Reviewer 2 Report
Comments and Suggestions for Authors
This revised draft is improved and most of my concerns have been met. There are still a few lingering issues that I feel could be addressed to strengthen the paper. See the attached document.

Author Response
We would like to extend our sincere appreciation to the reviewer for the meticulous and insightful comments during the second-round review. We are pleased to inform that all the recommendations have been incorporated into the revised manuscript. We believe that with these modifications, the paper now presents a more robust and refined contribution to the field.
Thank you for your invaluable guidance and support.
